# The Role of Transcranial Magnetic Stimulation and Peripheral Magnetic Field Therapy in Chemotherapy-Induced Peripheral Neuropathy: A Narrative Review

**DOI:** 10.3390/cancers17223628

**Published:** 2025-11-12

**Authors:** Elena Wernecke, Faten Ragaban, Peter B. Rosenquist, Nikhil Jaganathan, William J. Healy, Egidio Giacomo Del Fabbro

**Affiliations:** 1Medical College of Georgia, Augusta, GA 30912, USA; ewernecke@augusta.edu (E.W.); njaganathan@augusta.edu (N.J.); 2Division of Palliative Medicine, Medical College of Georgia, Augusta, GA 30912, USA; fragaban@augusta.edu; 3Department of Psychiatry and Health Behavior, Medical College of Georgia, Augusta, GA 30912, USA; prosenquist@augusta.edu; 4Division of Pulmonary, Critical Care, and Sleep Medicine, Medical College of Georgia, Augusta, GA 30912, USA; wihealy@augusta.edu

**Keywords:** chemotherapy-induced peripheral neuropathy, transcranial magnetic stimulation, repetitive transcranial magnetic stimulation, cancer symptoms

## Abstract

Chemotherapy-induced peripheral neuropathy (CIPN) often causes severe, persistent pain in patients undergoing chemotherapy. The current medication of choice, duloxetine, reduces pain in one out of every five patients treated. Opioids also provide partial relief but carry a risk of serious side effects. There is a need to develop effective treatments with fewer side effects. We review the results of trials for pain relief in CIPN, using magnetic fields to the brain or to the peripheral nerves in the limbs of patients. The initial results show benefit for pain relief, with few side effects.

## 1. Introduction

Chemotherapy-induced peripheral neuropathy affects 68% of patients within their first month of cancer treatment and significantly decreases quality of life [1,2,3,4]. Overall, up to 90% of patients treated with taxanes, platinum-based agents, vinca alkaloids, or proteasome inhibitors [1] experience neuropathic symptoms at some time. Up to 42% of patients report lingering neuropathy after discontinuation of treatment, while others may encounter worsening of pain for several months (referred to as coasting phenomenon) [3]. Debilitating symptoms frequently result in dose reduction or discontinuation of chemotherapeutics [4], which alters treatment plans, potentially impacts survival and may not offer relief [5]. In older patients, studies indicate CIPN may be particularly burdensome given its association with reduced executive function [6], cognitive decline [7], and more severe pain [8,9]. In addition, neuropathy symptoms in oncology patients are associated with living alone, increased comorbidities [10], and polypharmacy [11].

The American Society of Clinical Oncology (ASCO) and European Society for Medical Oncology (ESMO) guidelines for CIPN recognize duloxetine as the agent with the best evidence for established, painful CIPN although the benefit is noted as ‘limited’ [4,12]. ASCO guidelines also underscore the need to identify ‘better’ interventions for preventing and treating CIPN.

The use of magnetic fields to modulate central and peripheral neurons demonstrates potential for relieving neuropathic pain, with few adverse effects. The intervention may also mitigate associated symptoms of depression and anxiety which often occur as a symptom cluster with pain. Two broad intervention techniques are available: transcranial magnetic stimulation (TMS) and peripheral magnetic field therapy (MFT). Although there are no direct comparison studies, each of these modalities has potential advantages and disadvantages for managing pain and specifically CIPN. This review aims to contrast these modalities specifically in CIPN.

### 1.1. Transcranial Magnetic Stimulation (TMS)

TMS is a noninvasive method of neuronal stimulation via a wire coil placed against the target region, generating a magnetic field that passes through the overlying tissue, inducing electrical activity in neurons. In cortical regions such as the primary motor cortex (M1), stimulation engages cortical networks and activates the corticospinal tract and spinal motor neurons leading to contraction in the contralateral muscle. Motor-evoked potentials of the muscle can be measured by electromyography (EMG), indicative of the degree of cortical stimulation [7]. Various coil configurations placed over the scalp may differentially activate superficial or deeper cortical areas and can provide broader or more focused stimulation. As opposed to single pulse TMS, repetitive TMS (rTMS) can be used to produce long-term alterations of activity, possibly due to long-term potentiation or depression, although the exact mechanism is uncertain [7]. Inhibitory effects occur with low frequency stimulation of <1 Hertz (Hz), while excitatory effects occur with high frequencies > 5 Hz [13]. Intensity is determined on a case-to-case basis through the determination of the patient’s motor threshold (MT). Frequently, studies employ intensities of 80–120% of the patient’s MT [14].

### 1.2. Peripheral Magnetic Field Therapy (MFT/PEMF)

Peripheral magnetic field therapy (MFT), also termed pulsed electromagnetic field therapy (PEMF), refers to a non-invasive treatment with similar technology to rTMS. A magnetic field is generated and applied directly to the affected extremity, resulting in nerve excitation and muscle stimulation [15]. Dosing is determined by multiple parameters including flux density, frequency, and session duration. Currently, formal studies are lacking regarding the most effective intensity and frequency [16], but pain improvement is shown in both low (<0.5 Hz) and higher (>10 Hz) frequencies, depending on the pathophysiology of the pain syndrome being treated [17].

TMS suggests benefit in the short-term treatment of peripheral neuropathic pain, including non-cancer pain refractory to drug therapy [18]; in diabetic neuropathy [19,20]; and in other disorders responding to neuromodulation [21]. PEMF shows efficacy in modulation of inflammation, acceleration of wound healing, and in improving musculoskeletal-related disorders such as osteoarthritis or tendon injury [16]. This narrative review aims to consolidate information from recent studies describing the potential role of TMS and PEMF in cancer-related pain and more specifically in CIPN [22,23]. To our knowledge, no other reviews have directly assessed magnetic stimulation in the context of CIPN or malignancy-associated neuropathy.

## 2. Methodology

### 2.1. Search Strategy

A structured search of the literature using PubMed and the following keywords:•“Chemotherapy induced peripheral neuropathy” or “CIPN” AND “transcranial magnetic stimulation” OR “TMS”•Yielded 6 results: 2 quasi-experimental studies included, 2 systematic reviews excluded, 2 case reports excluded

The search was then broadened to include peripheral therapy with the following keywords:•“Chemotherapy induced peripheral neuropathy” OR “CIPN” AND “transcranial magnetic stimulation” OR “peripheral magnetic stimulation”•Yielded 157 results: 10 of these were screened for more thorough review, and 5 were included in the final review.

An additional search of the Ovid database using the following keywords:•“Chemotherapy induced peripheral neuropathy” OR “CIPN” AND “transcranial magnetic stimulation” OR “TMS” OR “peripheral magnetic stimulation”•Yielded 70 results: Title screening revealed one result that had already been included in the final review.

A diagram of the selection process has been included (Figure 1).

### 2.2. Inclusion and Exclusion Criteria

Studies included:•Publications in English.•TMS or PEMF as a potential treatment for peripheral neuropathy (including CIPN and other peripheral neuropathy in the setting of cancer)•Randomized controlled trials (RCTs), clinical trials, or observational studies.•Published within the last 10 years (2015–2025)

Studies excluded:•Exclusion of patients with cancer.•Case reports or review articles.

## 3. Results

### Included Studies

Seven articles were included in the review. Two studies directly addressed the use of rTMS for CIPN [24,25]. Two additional studies addressed the use of peripheral magnetic therapy in CIPN [26,27]. Three other studies did not exclude cancer-related neuropathy or included subsets of patients with CIPN [22,28,29]. The results of the studies in Table 1, Table 2 and Table 3 reflect the ability of rTMS and PEMF to improve key outcomes of CIPN with minimal side effects. Benefit occurred in rTMS even at the lowest tested frequency (5 Hz), lowest tested pulse rate (500/session), and shortest duration (4 days) [25]. Additional studies directly comparing frequency, intensity, pulse rate, duration/number of treatments, and follow-up could help identify which combination of factors works best for CIPN specifically.

## 4. Discussion

### 4.1. Role in Management of CIPN: Repetitive Transcranial Magnetic Stimulation

Two preliminary trials of rTMS in CIPN support its efficacy for symptom management. Quantitatively, rTMS was associated with faster sensory and motor conduction velocities [24]. Qualitatively, patients reported improvement of symptoms, including significantly decreased visual analog scores (VAS) for pain [24,25] and dysesthesia [25]. Unfortunately, neither study had a control group, and each had few patients enrolled. However, this is still valuable information, especially given that patient reported outcomes are highly important in improving quality of life and functionality.

A randomized sham-controlled trial assessed malignancy-associated neuropathic pain, including 18 patients whose neuropathy could potentially be attributed to CIPN.

This study yielded similar results with improvement in multiple symptom scales, including verbal descriptor scale (VDS), VAS, Leeds assessment of neuropathic symptoms and signs (LANSS), and Hamilton rating scale for depression (HAM-D). Notably, these changes were observed after several sessions of rTMS, and the effects lasted 15 days, but ended after 1-month follow-up [22].

### 4.2. Technique

Technical factors, including placement of magnetic stimulation to the cranium, appear to be pertinent. Possible locations for coil placement include the primary motor cortex (M1) or dorsolateral prefrontal cortex (DLPFC). A multi-center sham-controlled trial investigated the relative efficacy of these two locations to each other and to placebo [28]. Compared to sham-rTMS, M1rTMS reduced pain intensity and improved pain relief, sensory dimension of pain, and scores on Patient and Clinician Global Impression of Change (PGIC and CGIC). DLPFC-rTMS, meanwhile, was non-superior to sham in each of these parameters. A dose-dependent relationship was demonstrated as repeated sessions increased the difference between M1-rTMS and sham-rTMS groups. The numbers needed to treat (NNT) with M1-rTMS for >50% pain relief, >50% improvement in numeric rating scale, and moderate improvement of PGIC score, were 3.1, 7.7, and 5.9, respectively [28]. Additional pilot studies of non-CIPN neuropathic pain (not included in this review) have successfully targeted other regions of the brain such as the posterior superior insula (PSI) with deep brain stimulation [18]. While M1 was most commonly treated in the cited papers of this review and in one paper was shown to be superior to DLPFC treatment, additional direct comparisons to other treated brain areas would be beneficial.

Additional factors to consider include orientation of coil over the M1 area, frequency, intensity, and pulses per session. One pilot study randomized patients to posterior–anterior or lateral–medial orientations and found similar efficacy when applied at the same intensity (optimal at 90% of the MT) [25]. All other studies of rTMS and CIPN in this review used intensity of 80% MT [22,24,28,29]. Efficacy was shown across multiple frequencies, ranging from 5 to 20 Hz. A meta-analysis of 24 studies, including 7 with neuropathic pain of varying etiologies, found 20 Hz to be the optimal frequency for reducing chronic pain using rTMS [30]. Of note, pain and dysesthesia improved even at 5 Hz, a lower frequency than is typically used [25]. Pulses per session varied from 500 to 3000 pulses, and duration of treatment/number of sessions varied from 4 sessions to 30 sessions [22,24,28,29]. Although none of these parameters were directly compared within a single study, repetitive TMS shows promise across studies with different frequencies, intensities, pulses, and duration.

### 4.3. Role in Management of CIPN: Peripheral Magnetic Stimulation

Efficacy of peripheral magnetic stimulation in CIPN was assessed in a double-blind RCT, with 44 patients randomized to the intervention (n = 21) or placebo arm (n = 23). Nerve conduction studies revealed improvement in mean peroneal nerve conduction at three months among those treated with PEMF, and mean nerve conduction at this time point was significantly better than the placebo group [26]. Patient-reported scores for neurotoxicity and neuropathic pain were used as secondary endpoints, with significantly decreased Common Terminology Criteria for Adverse Events (CTCAE) scores among the PEMF group by end of study. Unfortunately, no significant improvement was demonstrated for neuropathic pain using the Pain Detect End Score. This study confirmed the results of a pilot trial by the same investigators in 2015, showing improved nerve conduction velocity (NCV) of the sural nerve. However, among 20 patients with CIPN of varying degrees, neuropathic pain was only mildly and non-significantly lessened despite significantly improved scores for sensory ataxia and sensory neuropathy at 4 weeks. The lack of significant patient-reported improvement may reflect an inadequate frequency (4–12 Hz) of PEMF delivery since evidence supports higher frequencies for neuropathic pain [27].

### 4.4. Prediction of Therapeutic Response

A sham-controlled, double-blind study attempted a predictive model for patients most likely to experience treatment response from rTMS. A prediction score using three baseline variables was found to be highly sensitive (85%) and specific (84%). Two variables were associated with less benefit from rTMS trial, including depression (scoring > 8/21 on Hospital Anxiety and Depression or HAD Scale) and presence of distal lower extremity pain. One variable, the “magnification” score on pain catastrophizing scale (PCS), was associated with improved outcomes. Using these variables, the authors developed an equation for a prediction score. Scores of two or greater are correlated with good response of CIPN symptoms to rTMS [29]. Additional studies correlated initial symptom severity or burden with response to therapy. One cohort study of 30 patients demonstrated that all patients with grade 2, 80% with grade 3, and 20% with grade 4 CIPN experienced improvements in their symptoms following rTMS, indicating that lower grades of CIPN burden at onset of therapy may be more likely to remit [24]. Meanwhile, a double-blinded randomized trial showed peripheral PEMF to be especially efficacious for more severe symptoms as evidenced by a stronger improvement of NCV in patients with very low initial NCV [26]. Therefore, rTMS and peripheral stimulation may vary in their efficacy for relieving severe CIPN. Comparison studies are warranted to identify the most optimal target population for these modalities.

### 4.5. Prediction, Prevention, and Progression to Chronic CIPN

Predisposing factors for developing CIPN include increased age, African American race, genetic factors such as single nucleotide polymorphisms, and behavioral factors such as sedentary lifestyle and sleep disturbance [1,31,32,33]. Specific health-related predisposing factors include previous chemotherapy, hereditary neuropathy, diabetic neuropathy, use of alcohol, and possibly obesity [1]. Some chemotherapeutic drugs such oxaliplatin and cisplatin are more likely to manifest with chronic CIPN, and also with ‘coasting’ phenomenon, especially in the upper extremities [34].

Unfortunately, there are currently no effective options for preventing CIPN development [4,35,36]. No studies in this review assessed prevention of CIPN, or progression from acute to chronic CIPN as outcome measures.

Although we identified no studies of rTMS predicting transition from acute to chronic CIPN, there is evidence for its role as a biomarker identifying increased pain sensitivity and a predictor of progression to chronic pain. A study of healthy adults with induced-temporomandibular pain found the combination of rTMS plus electroencephalography and machine learning could accurately predict increased pain among the participants. The authors hypothesized that the biomarker has ‘substantial potential’ for predicting transition from acute to chronic pain [31]. This predictive ability using corticomotor excitability and sensorimotor peak alpha frequency was replicated in a study by the same investigators [37]. Normal volunteers with nerve growth factor-induced temporomandibular joint pain underwent pretreatment with rTMS to the M1 region, resulting in lower pain intensity on chewing and yawning.

### 4.6. Safety

Given older patients with cancer have increased comorbidities and risk for polypharmacy, the safety profile of any CIPN intervention is especially important. In the studies of patients with cancer, the most common side effect was mild-moderate headache or migraine that resolved with non-opioid analgesics [18,28]. In non-cancer populations using rTMS, the most serious adverse effect was induction of seizures, though these are very rare [38,39].

A recent review of seizure events and rTMS use identified 41 total seizure events among 36 studies dating up to 2020 [40]. A survey-based study involving questionnaires sent to authors of TMS-related papers encompassing multiple coil types and frequencies, found < 1 seizure per 60,000 sessions [41]. Seizure risk was increased in patients with known seizure risk factors, and seizure occurred more commonly in TMS-naïve patients (within the first three sessions).

The assessment of risk associated with TMS differs somewhat between the United States (US) and European Union (EU). In 2022, the EU reclassified non-invasive brain stimulation, including rTMS, as class III devices, indicating a risk level akin to invasive techniques [42]. In the US, rTMS is considered a moderate-risk class II device, with regulations designed to ’mitigate the potential risks to health’. It is contraindicated in use with metal implanted devices such as pacemakers, implantable cardioverter defibrillators, pumps, or cochlear implants. The Food and Drug Administration has approved rTMS for use in major depressive disorder (2008) and OCD (2018) in the United States [40]. Unfortunately, patients may bear a financial burden, with estimates of up to $300 per session. TMS techniques typically involve a rigorous weekly regimen of treatments, compounded by additional clinic visits for cancer treatment. Lack of access to healthcare services and other barriers such as childcare, transportation, or time off from work can present major challenges. This affects translational outlook for this therapy.

### 4.7. Additive Benefits

Repetitive TMS and peripheral magnetic stimulation therapy for CIPN may improve other outcomes beyond neuropathic pain and improved quality of life. For example, TMS may be opioid sparing, given the reduction in pain scores and evidence for increasing endogenous opioids. In one study, concomitant intranasal naloxone administration diminished pain relief from TMS compared to patients administered an intranasal placebo. After metabolism of the naloxone, the analgesic effect of the TMS was restored [43]. Another study among postoperative patients showed TMS to the DLPFC region significantly decreased attempts for medication delivery through a patient-controlled analgesia pump compared to sham [44]. However, additional studies are needed to show a longer, sustained opioid sparing effect, specifically in CIPN.

Depression is prevalent among cancer survivors, estimated at 17% at 5–10 years [45]. Given a large body of work demonstrating efficacy of rTMS in depression [46], one might expect a concomitant benefit for patients with CIPN and mood symptoms. The concept of total pain, comprising physical, social, psychological, and spiritual components, typically requires a multimodal therapeutic approach [47]. These components influence each other and patients’ ability to cope with their pain. One study found that questionnaire-identified depression was less likely to be associated with good response to rTMS [29]. However, based on its efficacy for depression, it is reasonable to suppose TMS may prove to be useful in mitigating a symptom cluster that includes pain, depression, and existential suffering. This requires further investigation.

### 4.8. Limitations

Notably, the few studies evaluating rTMS or MFT in CIPN generally have small sample sizes and lack sham control. These factors decrease the generalizability of results and do not account for the placebo effect, which may be significant in cancer-related symptom research [48]. This review is also limited in that some papers were included which studied cancer-related neuropathy but not CIPN directly. This was done for completeness. Multi-center clinical trials may also be challenging, given the limited number of centers offering this therapy and its resource-intensive nature. Methods of assessing CIPN could also be standardized for consistency—various tools such as CTCAE, EORTC QLQ-CIPN20, VAS for pain, SFMPQ2, and PCS have all been used in trials. One commonly used questionnaire in clinical trials, but not demonstrated in this review, is the Functional Assessment of Cancer Therapy/Gynecologic Oncology Group–Neurotoxicity (FACT-GOG-NTX). Patient-reported symptoms of CIPN are more sensitive outcomes than clinician-based assessments of CIPN. However, a combination of patient-reported outcomes, objective measures, and clinical assessment might in future prove to be best at defining the phenotype. Sensory and motor nerve conduction outcomes provide additional evidence but are described as lacking sensitivity. One tool, the total neuropathy score, combines physical exam, patient reported measures, and NCV, and is proposed as a highly sensitive scale for assessing changes [49].

## 5. Conclusions

In addition to its therapeutic role in pain and symptom relief, TMS may prove useful in identifying a biomarker that predicts the development of chronic CIPN. While preliminary studies show potential for magnetic stimulation therapy in improving patient reported outcomes, robust trials are limited, lacking appropriate controls, sample sizes, or consistent methodologies. Future sham-controlled studies in patients with cancer are necessary to confirm the positive outcomes discussed in this review. Moreover, the safety profile of magnetic stimulation and the potential for tumor inhibition [50] suggests the intervention may gain wider adoption in future. In particular, older patients may benefit given the higher prevalence of CIPN [51], increased symptom burden [52], and risk of drug side effects from gabapentin and opioids [53].

## Figures and Tables

**Figure 1 cancers-17-03628-f001:**
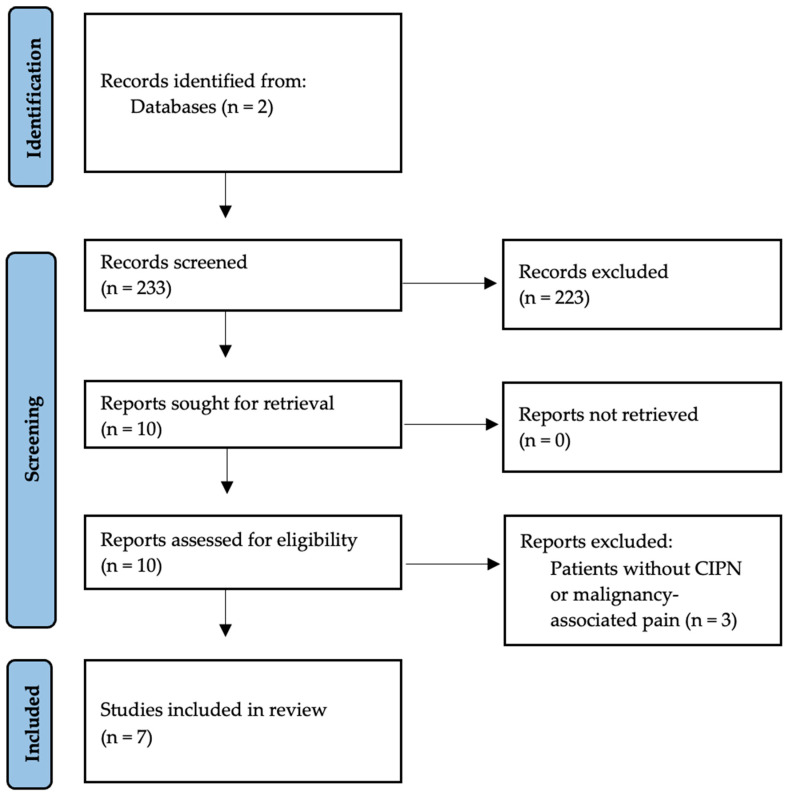
Selection of studies from identification to inclusion.

**Table 1 cancers-17-03628-t001:** Summary of seven studies describing the use of noninvasive magnetic therapies (rTMS or PEMF) in CIPN or malignancy-associated neuropathy.

Study—Authors and Date	Central vs. Peripheral	Frequency and Intensity	Total Pulses per Session	Interval	Sample Size, Description	Key Outcomes: Nerve Conduction Velocity (NCV)	Key Outcomes: Patient Reported Outcomes and Symptoms
Yan Z, Cao W, Miao L et al. Repetitive transcranial magnetic stimulation for chemotherapy-induced peripheral neuropathy in multiple myeloma: A pilot study. 2023 [24]	Central; Coil aligned with M1 region of brain.	10 Hz; 80% motor threshold	1400 pulses per session	5 sessions per week for 6 weeks	Retrospective cohort study of 30 multiple myeloma patients with CIPN. Pre- and post-treatment NCV, visual analog scale (VAS) for pain, and European Organisation for Research and Treatment of Cancer Quality of Life Questionnaire—Chemotherapy-Induced Peripheral Neuropathy (EORTC QLQ-CIPN20) were assessed. Median age: 52 years	Enhanced sensory and motor conduction velocity in peripheral nerves following rTMS.	Among patients who received rTMS, 80% experienced CIPN symptom reduction. Following rTMS, VAS for pain decreased.
Goto Y, Hosomi K, Shimokawa T et al. Pilot study of repetitive transcranial magnetic stimulation in patients with chemotherapy-induced peripheral neuropathy. 2020 [25]	Central; Figure-8 shaped coil to M1	5 Hz; 90–100% motor threshold	500 pulses per session	4 sessions within 2 months	Cohort study. 11 patients with breast cancer or gynecologic cancer with CIPN of severity ≥ 2. Pre-and post-treatment VAS of pain and dysesthesia and pain intensity reported through Short-form McGill Pain Questionnaire 2 (SFMPQ2). Mean age: 64.8 ± 7.8 years Age range: 55–77	Not applicable	Decreased VAS of pain and dysesthesia in the target extremity and decreased intensity of pain reported on the SFMPQ2.
Rick et al. Magnetic field therapy in patients with cytostatics-induced polyneuropathy: A prospective randomized placebo-controlled phase-III study. 2016 [26]	Peripheral; magnetic field therapy using MAGCELL device (PHYSIOMED ELEKTROMEDIZIN, Schnaittach, Germany). Affected palm or sole placed on the device.	4–12 Hz; 420 millitesla (mT) from device, measured strength of 105 mT		Twice daily over 5 min. Of note, Occupational therapy was also conducted 3 times per week.	Double-blind RCT. 44 patients: 21 in treatment group and 23 in placebo group. The primary endpoint was NCV, and secondary endpoints were Common Terminology Criteria for Adverse Events (CTCAE) score and Pain Detect End score. Median age: 58 years Age range (group 1): 28–73 years Age range (group 2): 43–73 years	Significant difference between control and study groups at the end of the study for peroneal nerve conduction.	CTCAE scores significantly reduced in both groups. Statistically significant difference between experimental and control group at T2 and T3. No difference in Pain Detect End Score versus placebo at endpoint.
Geiger et al. Low frequency magnetic field therapy in patients with cytostatic-induced polyneuropathy: a phase II pilot study. 2015 [27]	Peripheral; magnetic field therapy using MAGCELL device. Affected palm or sole placed on the device.	4–12 Hz; 420 mT from device, measured strength of 105 mT		Twice daily over 5 min.	Phase II pilot study. 20 patients with CIPN included, primarily with degree 1–2 CIPN. Median age: 59 years Age range: 42–73 years	Sural nerve exhibited significant increase in NCV.	Common Toxicity Criteria (CTC) scores for sensory ataxia and sensory neuropathy were significantly improved by the end for the study period. Neuropathic pain (scored by Pain Detect End questionnaire) exhibited a decrease in symptoms that was nonsignificant.
Attal, et al. Repetitive transcranial magnetic stimulation for neuropathic pain: a randomized multicentre sham-controlled trial. 2021 [28]	Central (M1 and DLPFC)	10 Hz; 80% of resting motor threshold	3000 pulses per session	15 total sessions spread out in increasing intervals throughout a period of 22 weeks.	Randomized multicenter sham-controlled trial. 149 patients aged 18–75 years with peripheral neuropathic pain treated with M1 rTMS, DLPFC-rTMS, or sham-rTMS over 25 weeks. 50 sensory polyneuropathy patients were identified including patients with CIPN. Mean age (M1rTMS): 56.7 ± 12.3 years Mean age (DLPFC-rTMS): 56.5 ± 11.6 years Mean age (Sham): 52.9 ± 11.9 years		M1rTMS reduced pain intensity and improved pain relief, sensory dimension of pain, PGIC and CGIC versus sham-rTMS. Meanwhile, DLPFC-rTMS was not superior to sham. Repeated sessions increased the difference between M1 and sham groups.
Attal N, Brander S, Pereira A, Bouhassira D. Prediction of the response to repetitive transcranial magnetic stimulation of the motor cortex in peripheral neuropathic pain and validation of a new algorithm. 2025 [29]	Central	10 Hz; 80% of resting motor threshold	3000 pulses/session	15 total sessions spread out in increasing intervals throughout a period of 22 weeks.	Secondary analysis of above RCT. 149 patients with peripheral neuropathic pain, including 50 with sensory polyneuropathy including CIPN. Mean age (M1rTMS): 56.7 ± 12.4 years Mean age (DLPFC-rTMS): 56.4 ± 12.1 years Mean age (Sham): 53.0 ± 12.1 years		Three baseline variables predicted sustained response to M1-rTMS with a Pearson score 0.58 (*p* < 0.001). Magnification score on pain catastrophizing scale (PCS) corresponded to increased response to rTMS. Presence of distal lower extremity pain and depression identified on HAD Scale predicted lesser therapeutic response. This algorithm demonstrated sensitivity of 85% and specificity of 84%.
Khedr EM, Kotb HI, Mostafa MG et al. Repetitive transcranial magnetic stimulation in neuropathic pain secondary to malignancy: a randomized clinical trial. 2015 [22]	Central, M1	20 Hz; 80% of motor threshold	2000 pulses/session	Daily for 10 consecutive days	Randomized, sham-controlled clinical trial; 34 patients with malignant peripheral neuropathy verified by validated tool, divided into two treatment arms (sham vs. treated with rTMS). VDS, VAS, LANSS, and HAM-D assessed throughout Mean age (Treatment): 47.0 ± 9.2 years Mean age (Sham): 48.0 ± 9.7 years	Not applicable	Improvement in all scales for up to 15 days. These improvements were not present 1 month later.

**Table 2 cancers-17-03628-t002:** Summary of effect sizes for key outcomes: nerve conduction velocity.

Study	Pre-Treatment	Post-Treatment	*p*-Value
* Yan et al. (2023) [24]	SCV (Sensory Conduction Velocity): 39.76 ± 4.93 m/s MCV (Motor Conduction Velocity): 43.83 ± 2.86 m/s	SCV: 44.29 ± 4.34 m/s MCV: 53.00 ± 2.19 m/s	SCV: *p* = 0.013 MCV: *p* = 0.002
Goto et al. (2020) [25]	Not assessed	Not assessed	Not assessed
^#^ Rick et al. (2016) [26]	SCV: 18 m/s	SCV: 40 m/s	Δ SCV after MFT: *p* = 0.006 MFT NCV > Placebo at 3 months: *p* = 0.015
^#^ Geiger et al. (2015) [27]	Sural nerve SCV: 24 m/s	Sural nerve SCV: 29 m/s	Δ Sural nerve SCV: *p* < 0.05
	Ulnar nerve SCV: 49 m/s	Ulnar nerve SCV: 52 m/s	Δ Ulnar nerve SCV: *p* > 0.05
Attal et al. (2021) [28]	Not assessed	Not assessed	Not assessed
Attal et al. (2025) [29]	Not assessed	Not assessed	Not assessed
Khedr et al. (2015) [22]	Not assessed	Not assessed	Not assessed

* SCV and MCV were compared on the left and right for multiple nerves (median, ulnar, peroneal, tibial). Only left median nerve changes are displayed in this table for the sake of conciseness. **^#^** Change in a value is indicated by a delta symbol, “Δ”.

**Table 3 cancers-17-03628-t003:** Summary of effect sizes for key outcomes: patient-reported outcomes.

Study	Pre-Treatment	Post-Treatment	*p*-Value
Yan et al. (2023) [24]	VAS: 5.40 ± 1.94 EORTC-QLQ-CIPN20: 17.68 ± 8.14	VAS: 3.10 ± 1.60 EORTC-QLQ-CIPN20: 10.50 ± 9.55	VAS for pain: *p* < 0.001 EORTC-QLQ-CIPN20: *p* < 0.001
* Goto et al. (2020) [25]	Pain VAS (LM): 55.3 ± 43.0 Dysesthesia VAS (PA): 73.6 ± 23.4 Dysesthesia VAS (LM): 73.3 ± 21.1 SF-MPQ2 (PA): 28.1 ± 27.3	Pain VAS (LM): 46.8 ± 38.6 Dysesthesia VAS (PA): 64.3 ± 28.3 Dysesthesia VAS (LM): 66.7 ± 29.2 SF-MPQ2 (PA): 23.5 ± 24.1	Pain VAS (LM): *p* = 0.03 Dysesthesia VAS (PA): *p* = 0.03 Dysesthesia VAS (LM): *p* = 0.04 SF-MPQ2 (PA): *p* = 0.01
^#^ Rick et al. (2016) [26]	CTCAE: 11 Pain Detect End Score: 16	CTCAE: 2 Pain Detect End Score: 6	Δ CTCAE at endpoint: *p* < 0.001 CTCAE: difference versus placebo at endpoint *p* = 0.04 Δ Pain Detect End Score: *p* = 0.001 No significant difference in Pain Detect End Score versus placebo *p* = 0.116
Geiger et al. (2015) [27]	Number of patients with grade 2–3 sensory ataxia on CTC: 18/20 (90%) Number of patients with grade 2–3 sensory neuropathy on CTC: 16/20 (80%) Median Pain Detect End Score: 13.7	Number of patients with grade 2–3 sensory ataxia on CTC: 7/20 (35%) Number of patients with grade 2–3 sensory neuropathy on CTC: 6/20 (30%) Median Pain Detect End Score: 11.8	Improved sensory ataxia: *p* = 0.0008 Improved sensory neuropathy: *p* = 0.003 Pain Detect End Score: *p* > 0.05
Attal et al. (2021) [28]		Adjusted effect estimate difference in Brief Pain Inventory (BPI) for the Group × Time interaction for M1-rTMS group: −0.048 ± 0.01; 95% CI: −0.09 to −0.01	*p* = 0.01
Attal et al. (2025) [29]	Key outcome was a predictive model of response to M1-rTMS	Predictive model sensitivity: 85% Predictive model specificity: 84%	Predictive model sensitivity: *p* = 0.005 Predictive model specificity: *p* < 0.0001
Khedr et al. (2015) [22]		Two-way ANOVAs with Time × Group interaction: VDS: F = 0.43, degrees of freedom (df) = 2.1(58.4) VAS: F = 8.07, df = 2.05(57.5) LANSS: F = 2.83, df = 2.5(70.4) HAM-D: F = 4.85, df = 2.37(66.3)	VDS: *p* = 0.0001 VAS: *p* = 0.001 LANSS: *p* = 0.018 HAM-D: *p* = 0.007

* Orientations of the coils were in either the lateral–medial (LM) or posterior–anterior (PA) orientation; VAS were collected regarding the target extremity as defined by the patient. **^#^** Change in a value is indicated by a delta symbol, “Δ”.

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
