# Peer review of "The Role of Transcranial Magnetic Stimulation and Peripheral Magnetic Field Therapy in Chemotherapy-Induced Peripheral Neuropathy: A Narrative Review"

_cancers, 2025, doi:10.3390/cancers17223628_

Round 1
Reviewer 1 Report
Comments and Suggestions for Authors
Manuscript ID: cancers-3888911.
The authors provide a relevant overview of the potential role of non-invasive magnetic stimulation therapies (rTMS and PEMF) in managing chemotherapy-induced peripheral neuropathy (CIPN). The topic is of significant clinical interest, but several areas require improvement .
Major Comments:
- The manuscript is described as a“narrative review,” but the presentation style resembles a systematic review. Please clarify the review type and ensure the structure and content are aligned accordingly. If it is indeed a narrative review, more critical synthesis and less descriptive summarization are expected.
- The search strategy is inadequately described and lacks the necessary detail for reproducibility. The use of only one database (PubMed), the absence of a defined search date range, and the lack of a clear protocol for study selectionare major weaknesses.
- Only 7 studies were included in the final review, some of which were not specifically focused on CIPN. Sample sizes were generally small.
- The included studies exhibit significant heterogeneity in stimulation parameters, treatment duration, and assessment tools. The authors did not systematically discuss how this heterogeneity impacts the interpretation of the results.
- The conclusion states "preliminary studies show promise" but fails to explicitly emphasize key limitations such as the preliminary nature of the evidence, lack of control groups, and small sample sizes.
Minor Comments:
- Please revise for language consistency and clarity. Some sentences are overly long or repetitive.
- Ensure all abbreviations are defined at first use and used consistently throughout.
- There are minor typographical errors (e.g., "DLPCF" should likely be "DLPFC")
Please revise for language consistency and clarity. Some sentences are overly long or repetitive.
Reviewer 2 Report
Comments and Suggestions for Authors
This review explores magnetic stimulation therapies (TMS and PEMF) as emerging, low-risk treatments for chemotherapy-induced peripheral neuropathy (CIPN).
Although based mainly on small-scale studies, the review contributes meaningfully to oncology, pain medicine, and neurorehabilitation by outlining future directions and emphasizing the need for standardized sham-controlled clinical trials.
Here some major comments - related to concepts and methdology.
- The paper woudl benefit of at least a graphical element. I would advise a PRISMA-like flow diagram of included studies would enhance comprehension and visual appeal.
- The auhors are quidly asked to clarify the research gap. At the end of introduction ould explicitly state the distinct contirbution between their paper and previous reviews, e.g. systematically contrasted both modalities (central and peripheral magnetic stimulation) in cancer-related neuropathy. This seems to be one major contribution but they have to support such a statement by referencing similar works reviewing either TMS or PEMF.
- Of course this is a narrative review, but adding a brief comparative summary table of effect magnitudes (VAS changes, NCV improvements, etc.) would make the results more compelling..
- Please check the uniform use of of acronyms and units (e.g., Hz, mT, NCV) throughout the text and tables.
- The purpuse of such a narrative paper is not only to cover or indicate a knowledge gap but to indicate translational outlook, clinical applicability, accessibility, and cost considerations. This is very importent to make the paper impactful for oncology practitioners.
Finally some minor comments related to style and editorial aspects.
1. It was a little bit frustrating that authors were not listed with their affiliations. Please provide a manuscript with this issue completely solved.
2. There are some occasional mid-word line splits likely due to format conversion.
3. In Discussion there are few very long sentences. As a general rule, revise the manuscript and split compound sentences over, let's say, 40 words.
4. Replace repetitive phrases such as “shows promise” with varied academic expressions (“demonstrates potential,” “suggests benefit,” etc.).
5. I think you did not followed the Cancers guideline or template for Table and the text under the table. Please check this.
6. Are there any recent reviews on neuromodulation in oncologic pain and magnetic field bioeffects (2023–2025) that can be further taken into consideration and cited within this manuscript?
Reviewer 3 Report
Comments and Suggestions for Authors Thank you for inviting me to review this interesting article. The review addresses a very important aspect of side effects of systemic treatment.The authors provide a very interesting perspective on the use of magnetic fields
and TMS. As they presented, the number of studies for cancer survivors is significantly limited.
This stems from the still unclear effects of magnetic fields on cancer cell metabolism. In my opinion, the authors failed to provide information on the type of cancer and its stage or progression of cancer disease,
or the time since the end of systemic treatment. This is important information regarding the use of magnetic fields on potentially
circulating cancer cells. It would also be worthwhile for the authors to present how magnetic fields affect cell metabolism and what
the biochemical effects might be, which I recommend adding to the review. Regarding side effects, it's worth mentioning contraindications to these forms of therapy (e.g., epilepsy, implanted pacemaker, etc.).
Furthermore, I congratulate the authors on the idea for this review.
